# Pyrethroid susceptibility reversal in *Aedes aegypti*: A longitudinal study in Tapachula, Mexico

**Patricia Penilla-Navarro[1], Francisco Solis-Santoyo[1], Alma Lopez-Solis[1], Americo D. Rodriguez[1], Farah Vera-Maloof[1], Saul Lozano[2], Elsa Contreras-Mejía[3], Geovanni Vázquez-Samayoa[3], Rene Torreblanca-Lopez[3], Rushika Perera[4], William C. Black IV[4], Karla Saavedra-Rodriguez[4]***

**1** Centro Regional de Investigación en Salud Pública, Instituto Nacional de Salud Pública, Tapachula, Chiapas, México, **2** Centers for Disease Control and Prevention, Arboviral Diseases Branch, Fort Collins, Colorado, **3** Jurisdiccion Sanitaria VII, Tapachula Chiapas, Antiguo Hospital General de Tapachula, Tapachula, Chiapas, Mexico, **4** Center for Vector-Borne Infectious Diseases, Colorado State University, 1685 Campus Delivery, Fort Collins, Colorado

* ksaavedr@colostate.edu

**Data Availability Statement:** The authors confirm that all data underlying the findings are fully available without restriction. All relevant data are

## Abstract

Pyrethroid resistance in *Aedes aegypti* has become widespread after almost two decades of frequent applications to reduce the transmission of mosquito-borne diseases. Because few insecticide classes are available for public health use, insecticide resistance management (IRM) is proposed as a strategy to retain their use. A key hypothesis of IRM assumes that negative fitness is associated with resistance, and when insecticides are removed from use, susceptibility is restored. In Tapachula, Mexico, pyrethroids (PYRs) were used exclusively by dengue control programs for 15 years, thereby contributing to selection for high PYR resistance in mosquitoes and failure in dengue control. In 2013, PYRs were replaced by organophosphates—insecticides from a class with a different mode of action. To test the hypothesis that PYR resistance is reversed in the absence of PYRs, we monitored *Ae. aegypti*'s PYR resistance from 2016 to 2021 in Tapachula. We observed significant declining rates in the lethal concentration 50 ($LC_{50}$), for permethrin and deltamethrin. For each month following the discontinuation of PYR use by vector control programs, we observed increases in the odds of mosquitoes dying by 1.5% and 8.4% for permethrin and deltamethrin, respectively. Also, knockdown-resistance mutations (*kdr*) in the voltage-gated sodium channel explained the variation in the permethrin $LC_{50}$s, whereas variation in the deltamethrin $LC_{50}$s was only explained by time. This trend was rapidly offset by application of a mixture of neonicotinoid and PYRs by vector control programs. Our results suggest that IRM strategies can be used to reverse PYR resistance in *Ae. aegypti*; however, long-term commitment by operational and community programs will be required for success.

within the paper and its Supporting Information files.

**Funding:** Funding was provided to KSR, WCBIV, PPN, and ADR by the National Institute of Allergy and Infectious Diseases of the National Institutes of Health (https://www.niaid.nih.gov/) under Award Number R01AI121211 ("Insecticide Resistance Management to Preserve Pyrethroid Susceptibility in Aedes aegypti").The content is solely the responsibility of the authors and does not necessarily represent the official views of the National Institutes of Health. The funders had no role in study design, data collection and analysis, decision to publish, or preparation of the manuscript.

**Competing interests:** The authors have declared that no competing interests exist.

## Author summary

The mosquito *Aedes aegypti* is the principal urban vector of the viruses that cause three globally significant diseases: dengue fever, chikungunya, and Zika fever, for which vaccines and effective treatments are currently absent. The only way to control dengue, chikungunya, and Zika fever outbreaks is to diminish vector populations. During epidemics, the most frequent way of targeting adult mosquitoes is outdoor spatial spraying of insecticides. Control of *Ae. aegypti* is difficult because limited insecticide classes are available for public health, leading to operational practices that overuse single molecules for long periods of time, imposing great selection pressure on mosquito populations for insecticide resistance. Insecticide resistance management (IRM) is proposed as a strategy to prevent resistance and avoid depleting the susceptibility resource in mosquito populations. IRM strategies assume that alternation of insecticides with different toxicological modes of action will prevent the selection of resistance. Unfortunately, very few field evaluations have reported IRM schemes to control *Ae. aegypti*. In our study, the exclusive use of pyrethroids in vector control programs in Mexico from 1999 to 2013, led in the selection of knockdown resistance (*kdr*) to pyrethroid insecticides. To address this issue, vector control programs temporarily phased out pyrethroids from 2013 to 2019, substituting them with an insecticide class with a different mode of action: organophosphates (OPs). During six years, we monitored pyrethroid resistance in 24 mosquito populations from Tapachula, Mexico. We show that discontinuing pyrethroids for six years led in pyrethroid resistance reversal in *Ae. aegypti* in the field. However, high levels of pyrethroid resistance continue to jeopardize operational application, necessitating longer periods of pyrethroid cessation and novel IRM strategies to achieve lower resistance thresholds.

## Introduction

Insecticide spraying is an important tool for mitigating *Aedes* adult populations and reducing the transmission of arboviruses such as dengue, Zika, and chikungunya [1]. The most used adulticides for *Aedes* control programs are classified by their mode of action as sodium channel modulators (e.g., pyrethroids and DDT) or acetylcholinesterase inhibitors (e.g., organophosphates and carbamates) [2,3]. Because few formulations are approved and available for public health use, the operational tendency in dengue vector control is the use of a chemical class for long periods, resulting in an intense selection for resistance that eventually leads to mosquito control failure. Then, there is a subsequent switch to an alternative insecticide until resistance develops for that insecticide resulting in failure again. To resolve this challenge, insecticide resistance management (IRM) has been proposed as a strategy to retain the use of insecticides on natural populations [4].

IRM involves temporal or spatial rotations among alternative insecticides with different modes of action [3]. Ideally, these schemes reduce selection pressure from any one insecticide for any resistance mechanism [5–7]. Strikingly, few studies have demonstrated the value of IRM schemes in mosquito populations. One of these studies reported that annual rotations and spatial mosaics of pyrethroids (PYR), organophosphates (OP), and carbamates (CARB) reduced PYR resistance in *Anopheles albimanus* in Mexico compared to an exclusive PYR scheme [8,9]. In *Ae. aegypti*, a 3-month randomized trial in Mexico showed high efficacy of indoor residual spraying of bendiocarb (CARB) compared to deltamethrin (PYR) in a region with widespread PYR resistance [10]. More recently, the impact of varying operational

treatment regimens was evaluated in PYR-resistant *Ae. aegypti* populations from Florida over 2.5 years, where PYR susceptibility did not decline [11].

In Mexico, PYR application by dengue control programs for nearly two decades resulted in widespread resistance in *Ae. aegypti* populations [12–14]. From 1999 to 2010, PYR type-1 formulations containing permethrin or phenothrin were used intensively. Then, from 2010 to 2013, PYRs type-2, which contain an alpha-cyano moiety, were used briefly. Both PYRs type-1 and type-2 target the voltage-gated sodium channel (VGSC); however, PYR type-2 increased toxicity is due to differences in the way the PYR binds to the VGSC and possibly to additional ion channels [15]. Despite this, two major resistance mechanisms—target site insensitivity and enhanced metabolism—confer resistance to both type-1 and type-2 pyrethroids in *Ae. aegypti* populations in Mexico [12–14]. Target site resistance is associated with amino acid replacements that prevent the PYR from binding to the VGSC, commonly referred to as knockdown resistance mutations (*kdr*) [16–18]. In Mexico, three amino acid replacements—V410L, V1016I, and F1534C—are associated with different levels of pyrethroid resistance [19–22]. The frequency of the three *kdr* mutations increased simultaneously from 2000 to 2013—a period of exclusive PYR use by vector control programs [23]. In addition, enhanced metabolism has been associated with PYR resistance in Mexican *Ae. aegypti* through bioassays and transcription analysis of insecticide detoxification-associated genes such as the carboxyl/cholinesterases (CCE), glutathione-s-transferases (GST), and cytochrome $P^{450}$ monooxygenases (CYP) [24,25].

Mechanisms of resistance, such as *kdr*-conferring mutations in *Ae. aegypti* commonly have been associated with a lower fitness in the absence of insecticides [26–29]. For example, two PYR-resistant laboratory strains from Thailand and Brazil were restored almost to susceptibility after 12 and 15 generations, respectively, without insecticide pressure [26,27]. Then a semi-field study showed that a PYR-resistant strain from Mexico lost phenotypic resistance in ten generations, whereas *kdr* frequencies were unchanged [28]. Recently, the absence of insecticides for eight generations resulted in a decline in PYR resistance and *kdr*-allele frequencies in *Ae. aegypti* from different geographical regions in Mexico [29]. These observations support the potential use of IRM schemes under the assumption that susceptibility alleles will replace resistance-conferring alleles when an insecticide is discontinued.

A major goal in this field study is to test whether PYR discontinuation and replacement with alternative chemicals cause the reversal of PYR resistance and *kdr*-conferring mutations in mosquito populations. We conducted a 5-year longitudinal observational study in which the vector control program in Tapachula, Mexico, replaced the use of PYRs with OPs from 2013 to 2019. We surveyed 24 mosquito sites in Tapachula and determined the $LC_{50}$ for permethrin (type-1) and deltamethrin (type-2) and *kdr* allele frequencies every year from 2016 to 2020. Our results show large spatial heterogeneity in PYR resistance across Tapachula. Despite the heavy use of OPs and a lack of control over the domestic use of PYR at our study site, we report a substantial decline in the pyrethroid $LC_{50}$s and one of the *kdr* alleles over time. Strikingly, the decline in $LC_{50}$s was rapidly offset by the reintroduction of PYRs in combination with a neonicotinoid. Because few insecticide classes are available for public health use, the evaluation of IRM schemes to maintain insecticide effectiveness is imperative.

## Methods

### Experimental design

Our hypothesis is that discontinuation of pyrethroids from vector control programs causes a decline in pyrethroid resistance in mosquito populations. From 2016 to 2020, we conducted a longitudinal study comprising mosquito collections in 24 geographical areas in Tapachula,

Mexico. Insecticide bioassays in the laboratory were used to assess the change in susceptibility to pyrethroids in these mosquito collections throughout time. Our generalized logistic model included the 'odds of dying' as a response to several factors in our bioassays, including insecticide concentration, collection site, and collection period (months after pyrethroid removal).

## Study site

The city of Tapachula in the state of Chiapas is located in southeastern Mexico near the border with Guatemala. Tapachula is located 177 m above sea level and has a population of 305,766 inhabitants within 303 km$^2$. The city has a tropical climate, with an average maximum temperature of 33˚C, a minimum temperature of 23˚C and an average annual precipitation of 1,399.5 mm. Tapachula has two very distinct seasons: the dry season from November to April and the rainy season from May to October.

## Insecticide applications by vector control programs

Tapachula's Vector Control District VII maintains detailed documentation of the insecticide use since 2011. Between 2011 and 2020, insecticide applications were conducted as whole-city or focalized spraying operations. Because Tapachula is a dengue endemic city, whole-city spraying is planned on a yearly basis to reduce mosquito populations from April to November. Every year, four application cycles consisting of four weekly sequential applications cover the city using ultra-low volume spraying (ULV). In contrast, focalized spraying is a reactive strategy to treat households with a "suspected" dengue case. Within a week of the report, dengue "suspected" households are treated with indoor residual spraying and/or intradomicile spatial spraying, and neighboring blocks are treated with ULV. The insecticide applications follow the national guidelines published by Centro Nacional de Prevention y Control de Enfermedades in Mexico (CENAPRECE) [30]. The data was used to summarize the total area treated by each insecticide formulation each year.

## Collection sites

We conducted a preliminary collection at 16 sites in 2016. These included 6 cemeteries ('panteon') located in towns alongside the Coast of Chiapas and 10 sites located in the city of Tapachula. A team of 10 people walked throughout the cemetery checking for larval breeding sites, mostly flower vases and water-holding containers scattered across the cemetery. Similarly, the preliminary collection in the city of Tapachula consisted in visual inspection of larval breeding sites in the patios of ~ 20 houses from each collection site. Larvae and pupae were transferred to plastic bags and then transported to the insectary at Centro Regional de Investigación en Salud Pública (CRISP). Adults were identified following emergence, and *Ae. aegypti* were placed in cages (30 cm$^3$) with other mosquitoes collected from the same site. We only used collection sites that provided ~500 *Ae. aegypti* females. The females were blood feed on a rabbit following the guidelines of the Centro Nacional de Programas Preventivos y Control de Enfermedades (CENAPRECE) approved by the Ethical Commission of the Instituto Nacional de Salud Pública (CENAPRECE guidelines), which allowed us to obtain sufficient $F_1$ and $F_2$ offspring for the insecticide bioassays. Environmental conditions consisted of 27 ± 2 ˚˚C temperature, 70–80% humidity, and a 12:12 h photoperiod.

After this initial collection (2016), we limited our study to the city of Tapachula. We randomly selected 24 sites in Tapachula for our longitudinal study. Larva was recurrently collected from these 24 sites at five discrete periods between 2018 and 2020. The sites were located in each of four quadrants: Northwest (NW), Northeast (NE), Southwest (SW), and Southeast (SE) (Fig 1). Each collection site consisted of nine consecutive blocks (3 blocks x 3 blocks x 3

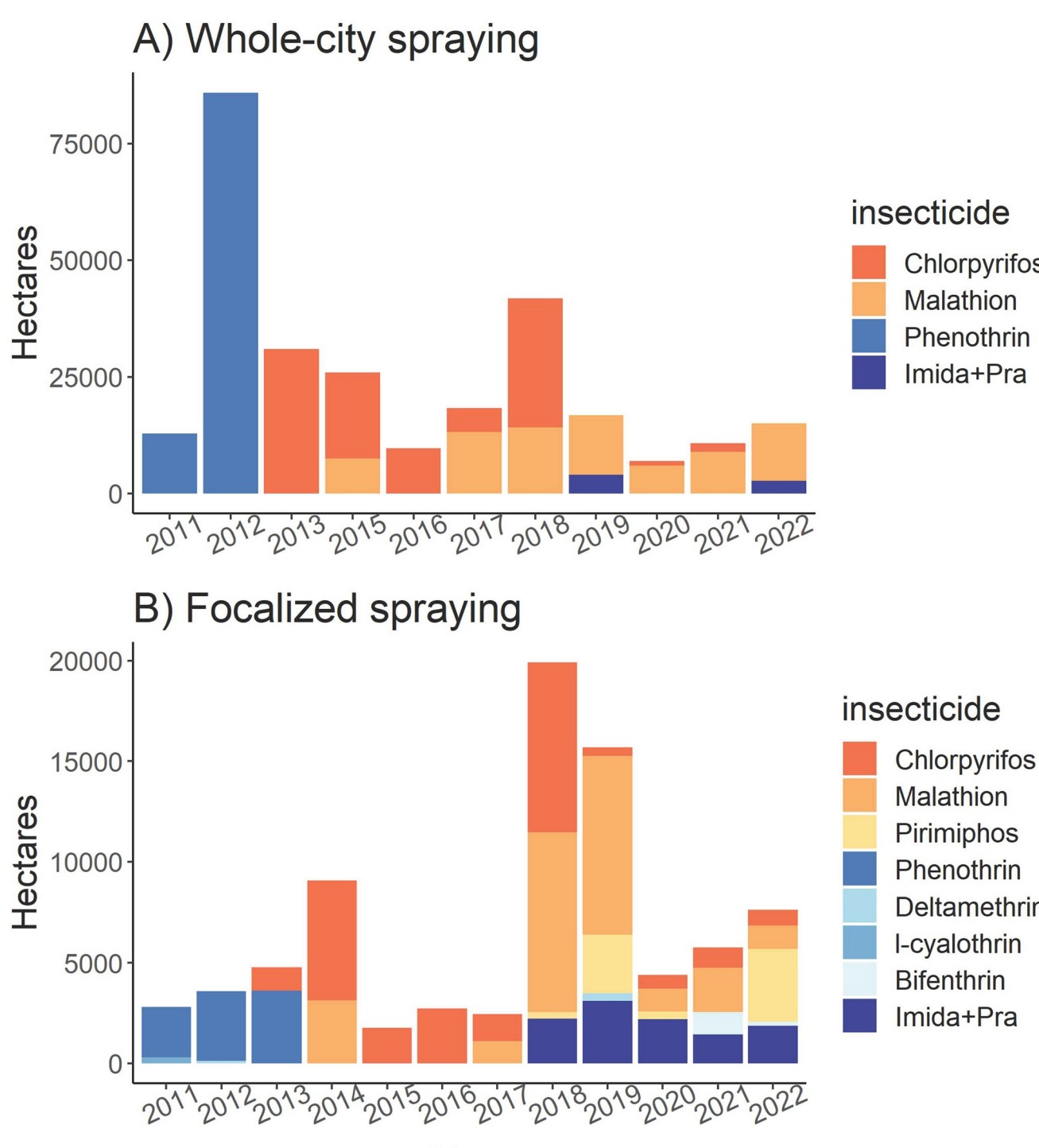

**Fig 1. Number of treated hectares during insecticide interventions in Tapachula from 2011 to 2022: A) Whole-city spraying and, B) Focalized spraying.** Data provided by Vector Control District VII in Tapachula, Chiapas. Imida + Pra = mixture of imidacloprid (neonicotinoid) and prallethrin (PYR type 1).

blocks). For the remaining of the study we used larvitraps (half of a rubber tire) filled with 3 L of water and suspended from a fence, window, or tree in the front or back yard of individual houses [31]. This collection method proved to be 3.6-fold more efficient than ovitraps to collect *Ae. aegypti* [31]. The water from the larvitraps was poured into plastic containers and transferred to the insectary; the larvitrap was then refilled with 3 L of water. Six houses were included per site, and sites were visited every week for 3 to 4 consecutive weeks or until 500–800 *Ae. aegypti* females per site were obtained.

## Collection times

At each site, we conducted five collections between 2018 and 2020. Additionally, collections were made at 17 sites in 2016 as a preliminary survey of PYR resistance levels and *kdr* alleles (Fig 2). Because pyrethroids were discontinued by vector control programs in May 2013, we timed each collection according to the number of months after pyrethroid discontinuation. The chronological collections were conducted in September 2016 (38 months after PYR removal), September 2018 (62 months), March 2019 (68 months), September 2019 (74 months), March 2020 (80 months), and September 2020 (86 months).

## Bioassays

The $F_1$ or $F_2$ adult offspring were exposed to insecticide concentrations in micrograms (μg/bottle) using the bottle bioassay procedures described in Solis-Santoyo et al. (2021) [32]. Between five and six insecticide concentrations were tested in triplicate. The technical grade insecticides (Supelco) permethrin (PYR type-1) and deltamethrin (PYR type-2) were used to coat 250 ml Wheaton bottles. Insecticide concentrations ranged between 10 μg/bottle and 160 μg/bottle for permethrin, and between 0.5 μg /bottle and 20 μg/bottle for deltamethrin. During the bioassay, 20 to 25 females (2–3 days old) were gently aspirated into each bottle. After 1 h of exposure, the mosquitoes were transferred to plastic containers and maintained in the insectary to observe the mortality at 24 h. The mortality caused by the different insecticide concentrations was used to calculate the lethal concentration 50 ($LC_{50}$), which is the amount required to kill 50% of the mosquitoes. Between 350 and 500 mosquitoes from each collection site were used to calculate the $LC_{50}$ for permethrin and deltamethrin. The $LC_{50}$ was also determined for the New Orleans (NO) susceptible reference strain every time we evaluated our collection sites (~4 sites evaluated per day) to confirm insecticide stock toxicity and to determine the resistance ratio (RR). For NO, concentrations ranged between 0.1 μg/bottle and 3 μg/bottle for permethrin and between 0.05 μg/bottle and 2 ug/bottle for deltamethrin.

The $LC_{50}$, 95% confidence intervals, slope, and intercept were determined using the binary logistic regression model with QCal software [33]. Then, a Pearson goodness of fit tested the adjustment of our data to the binary logistic regression model. The null hypothesis (Ho) assumed the observed mortality curve adjusts to a binary logistic regression model. Thus, we expected $p$ values greater than 0.05 to accept the Ho. When the Ho was rejected, the bioassay was excluded and repeated. Our study includes data from bioassays that adjusted to the binary logistic regression model ($p$ values > 0.05). To estimate the level of resistance among sites, we calculated a resistance ratio (RR) by dividing the $LC_{50}$ of the field sites by the $LC_{50}$ of the NO strain calculated during the day of evaluation.

## Genotyping *kdr* mutations

Genomic DNA was isolated from 50 individual $F_1$ female mosquitoes from each collection site following the procedures of Black and DuTeau [34]. The DNA was suspended in TE buffer (10 mM Tris-HCl, 1 mM EDTA pH 8) and stored at -20˚C. The V1016I and F1534C mutations

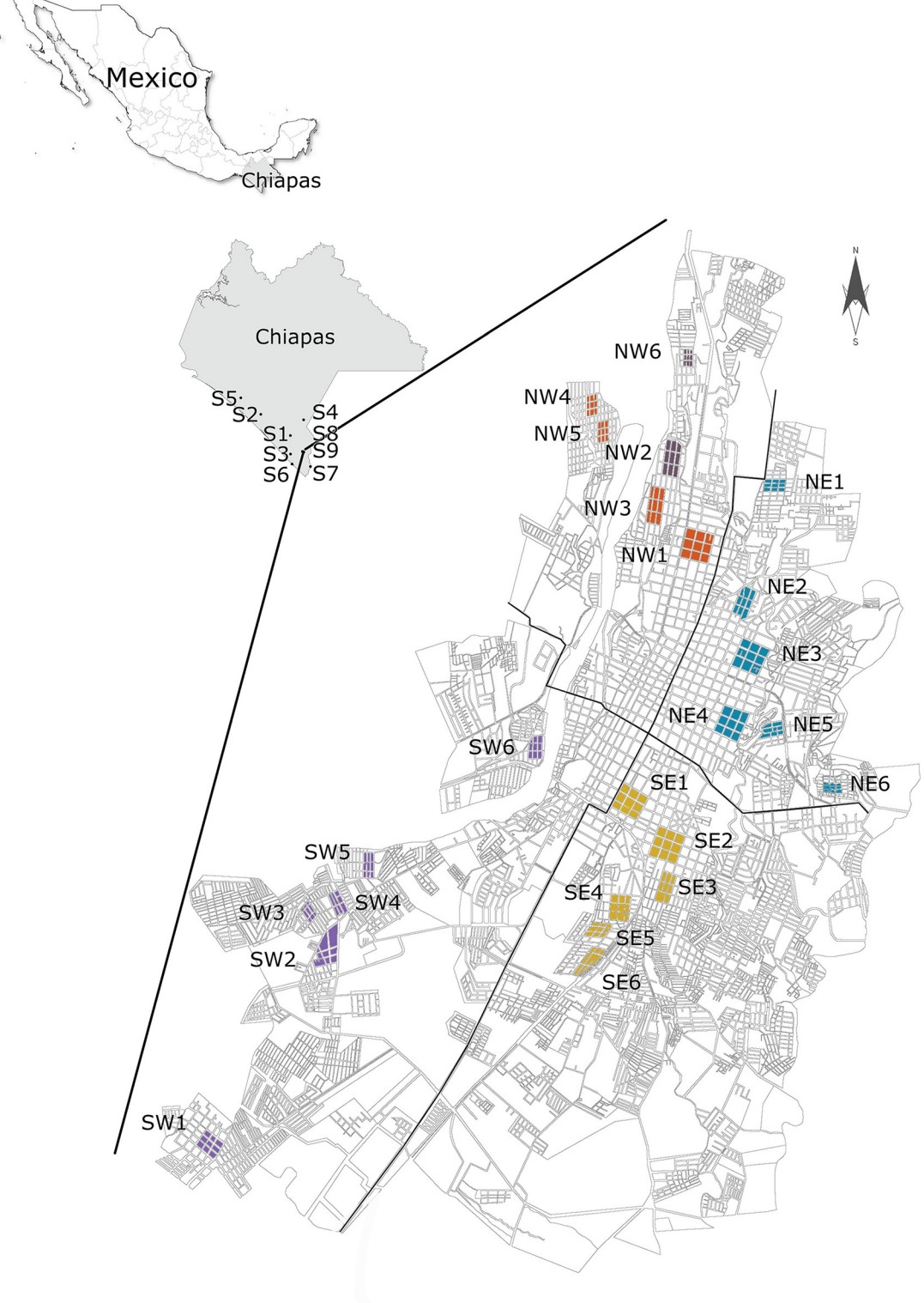

**Fig 2. *Aedes aegypti* collection sites in Tapachula, Chiapas, from 2018 to 2021.** Larvitraps were used to collect 500–800 females per site to generate the $F_1$ offspring for insecticide bioassays and kdr genotyping. Cemetery collections across the Coast of Chiapas were conducted in 2016. s1 = Huixtla, s2 = Mapastepec, s3 = Mazatan, s4 = Motozintla, s5 = Pijijiapan, s6 = Puerto Madero, s7 = Cd Hidalgo, s8 = Jardin, and s9 = Municipal Tapachula. Source: INEGI, Marco Geostadistico, 2022, Mexico (https://www.inegi.org.mx/temas/mg/#mapas and http://gaia.inegi.org.mx/mdm6). Other layers were obtained for U.S. Geological Survey (USGS) (https://www.usgs.gov/products/maps).

located in the VGSC gene were genotyped in 48 to 50 individuals per collection according to established protocols [19,21]. Between 714 and 1,200 individual mosquitoes were genotyped at each collection time point. In total, 5,616 individual mosquitoes were genotyped at both loci during the study period (2016 to 2020). The genotype frequencies at each locus were tested for Hardy-Weinberg (HW) equilibrium. The null hypothesis is that equilibrium is present in the population, which was verified with a chi-square test (df = 1 and *p* value > 0.05). Bayesian 95% Highest Density Intervals (HDI) around frequencies at loci 1016 and 1534 were calculated in WinBUGS for each collection site [35].

## Linkage disequilibrium between alleles at loci 1016 and 1534

Four potential 1016/1534 dilocus haplotypes were analyzed: $V_{1016}/F_{1534}$ (VF), $V_{1016}/C_{1534}$ (VC), $I_{1016}/F_{1534}$ (IF), $I_{1016}/C_{1534}$ (IC). The number of times (Tij) that an allele at locus *i* (1016) appears with an allele at locus *j* (1534) and an unbiased estimate of the composite disequilibrium coefficient ($\Delta ij$) were calculated using the program LINKDIS according the procedures in Vera-Maloof (2015) [17]. A $\chi^2$ test was performed to determine if significant disequilibrium exists among all alleles at loci 1016 and 1534. The statistic was calculated and summed over all two-allele interactions. The linkage disequilibrium correlation coefficient R*ij* is distributed from -1 (both mutations trans) to 0 (the 1534 and 1016 alleles occur independently) to 1 (both mutations cis) and therefore provides a standardized measure of disequilibrium.

## Generalized logistic regression

The bioassay data was subjected to a multivariate logistic regression model to test the relation between the odds of dying (dead/batch) and three independent variables: concentration of insecticide, time (in months after pyrethroid discontinuation), and collection site. The data were obtained from the bottle bioassays in which groups of mosquitoes (batch) were exposed to different concentrations of permethrin and deltamethrin for 1 h, and mortalities were scored at 24 h. The independent variable was the number of dead mosquitoes recorded at 24 h. The experimental unit was a batch of mosquitoes from a "site," collected in a "month," and exposed to a specific concentration of insecticide (µg/bottle). The 24 collection sites across Tapachula were included as nominal variables in the model. Time in "months" was included as numerical variables consisting of the number of months after the discontinuation of pyrethroids by vector control programs. Because, pyrethroid use was reestablished at the end of 2019 (as a mixture of imidacloprid and prallethrin), the regression analysis excluded the 80- and 86-month time points. The null hypothesis is that the odds of dying is independent of the insecticide concentration, time, and site. The generalized multinomial linear regression model was conducted using the MASS package in R (glm (dead/batch ~ dose + time + site)), using a binomial distribution.

## Results

### Insecticide use by vector control programs in Tapachula

The goal of our study is to evaluate the changes in pyrethroid resistance in mosquito populations as a response to pyrethroid discontinuation from vector control programs. Our study measured the changes in resistance by comparing the $LC_{50}$ between 2016 and 2020 across Tapachula, however, removal of pyrethroid occurred in 2013. In this section, we present the use of insecticides by vector control programs in the City of Tapachula from 2012 to 2020 with the goal of establishing the insecticide pressure over mosquito populations before and during

our study. From 1999 to 2010, permethrin synergized with piperonyl butoxide (PBO) was used exclusively by vector control programs [13]. Since 2011 CENAPRECE added classes of insecticides with a different mode of action, such as OPs and CARBs [30] to the list of approved insecticides. Tapachula's Vector Control District VII replaced the use of PYR with OPs in May 2013. The number of treated hectares in Tapachula's whole-city and focalized application is shown in Fig 1A and 1B, respectively. In 2012, whole-city spraying applied phenothrin (PYR) across 83,000 hectares. Then, between 2013 and 2022, OPs such as chlorpyrifos-ethyl and malathion were used intensively. In 2019 and 2022, primarily OPs were used; however, applications of OPs were alternated with a mixture of imidacloprid (neonicotinoid) and prallethrin (PYR). In contrast, the number of hectares treated with focalized spraying was three-fold lower than whole-city spraying (Fig 1B). Phenothrin was used until May 2013; in our study, this time is considered month 0 after the discontinuation of PYR. From 2013 to 2018, the focalized use of malathion, chlorpyrifos-ethyl, and pirimi-phos-methyl was reported. From 2018 to 2022, these OPs were alternated with a mixture of imidacloprid and prallethrin. Between 2020 and 2022, this mixture was used to treat between 25 and 60% of the focalized interventions. Additional insecticides used for indoor residual spraying included propoxur and bendiocarb in 2018; subsequently, bendiocarb was used exclusively in 2019 and 2020.

## Pyrethroid resistance in *Aedes aegypti* from Tapachula

We determined the levels of resistance to permethrin (PYR type-1) and deltamethrin (PYR type-2) in adult females using a customized bottle bioassay. Briefly, we exposed mosquitoes to five to six pyrethroid concentrations for 1 h, then transferred mosquitoes to observation cups and recorded mortality at 24 h. An initial screening in 2016 included ten sites in Tapachula and nine sites across the coast of Chiapas (Fig 2 and S1 Table). From 2018 to 2021, we conducted recurrent collections at 24 sites in Tapachula (Fig 2). The collections were conducted chronologically in September 2016 (38 months after PYR discontinuation), September 2018 (62 months), March 2019 (68 months), September 2019 (74 months), March 2020 (80 months), and September 2020 (86 months). The $LC_{50}$ of each collection site was calculated using a binary logistic regression model and then compared to the New Orleans (NO) susceptible reference $LC_{50}$ to determine the relative resistance ratios (RR). The permethrin and deltamethrin RRs calculated for each collection site between 2016 and 2020 are shown in Fig 3A and 3B.

**Permethrin resistance levels.**   In our initial screening in 2016, permethrin RRs among collection sites ranged from 10- to 81-fold. The RRs and their 95% confidence intervals for permethrin and deltamethrin at each collection site and time are shown in Fig 3A. In 2018, the RRs among collection sites ranged between 7- and 63-fold [32]. In March 2019 and September 2019, the RRs ranged from 7- to 52-fold and from 8- to 42-fold, respectively (Fig 3A). In March 2020, the RRs slightly increased from 22- to 59-fold after PYR was reintroduced by the vector control program in a mixture of imidacloprid and prallethrin (a neonicotinoid and a PYR type-1).

**Deltamethrin resistance levels.**   In 2016, RRs ranged between 21- and 190-fold (Fig 3B). In 2018 and March of 2019, RRs ranged between 11- and 102-fold and 8- and 81-fold, respectively. In September of 2019, we observed the smallest RR range, between 6- and 49-fold, suggesting that the removal of PYR from vector control programs resulted in deltamethrin resistance reversal. However, the RRs observed in March and September of 2020 increased to 22- and 127-fold and 22- and 88-fold, respectively. These increases corresponded to the reintroduction of PYR in a mixture of imidacloprid and prallethrin in 2019.

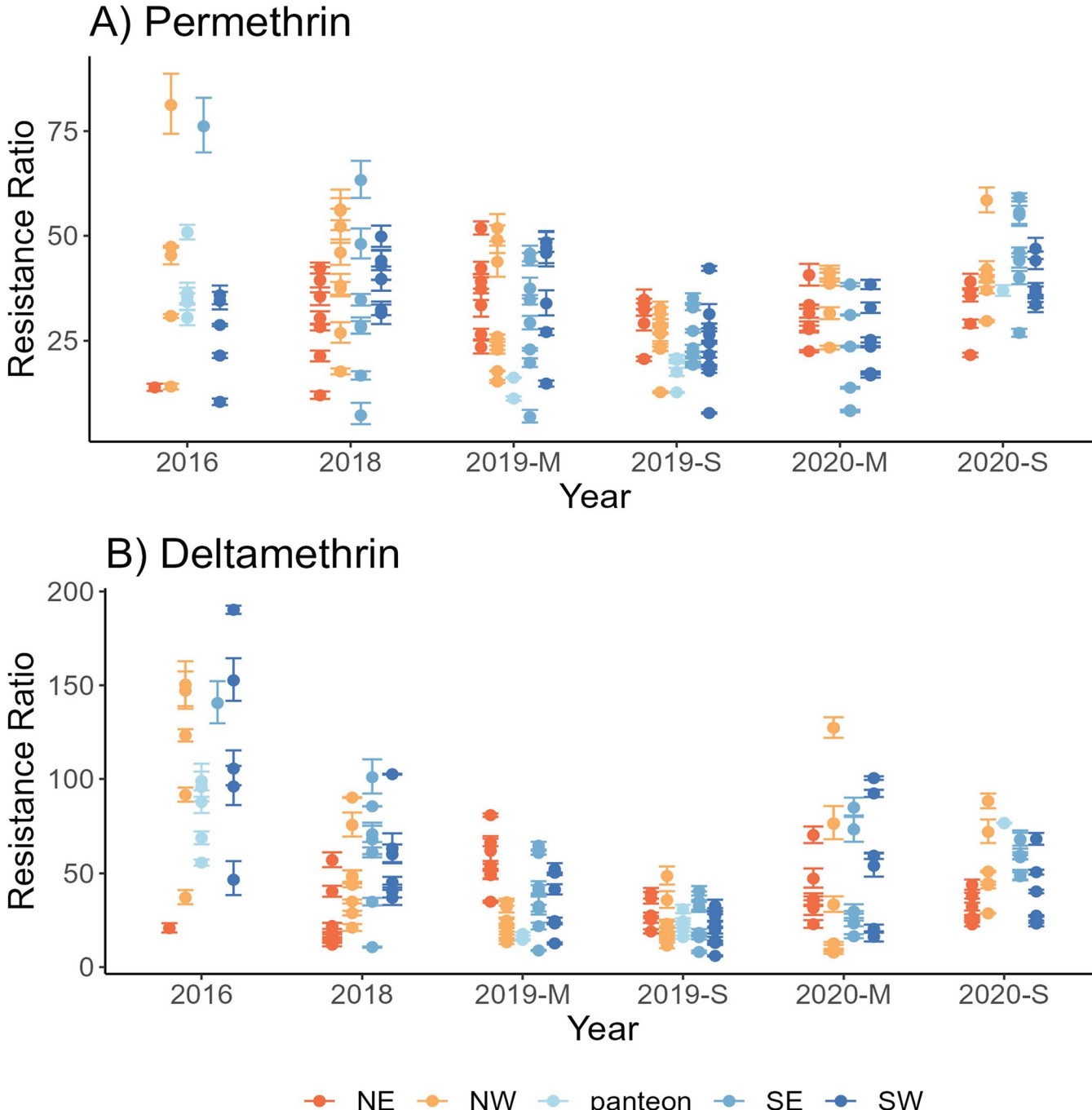

**Fig 3. Pyrethroid resistance ratios in *Aedes aegypti* collection sites from 2016 to 2020:** A) Permethrin and, B) Deltamethrin. Bars represent the 95% confidence intervals. Non-overlapping bars indicate the RR's were significantly different. None of our colonies were equal to the New Orleans reference strain RR. Sites are categorized within four geographical quadrants in Tapachula (NW = northwest, NE = northeast, SW = southwest, and SE = southeast). Collections from across coastal Chiapas were obtained from cemeteries (panteon). M = March and S = September. Resistance ratios are relative to the New Orleans $LC_{50}$ calculated at each time point.

**Modeling the odds of dying according to the site, time from PYR-removal, and insecticide concentration.** We conducted a multivariate logistic regression to test the relation between the odds of dying and three independent variables: site, time in months, and

insecticide concentration. The bioassays provided the number of dead mosquitoes out of the total mosquitoes exposed to a specific concentration of insecticide (dose-response). This strategy allowed us to include the dose-response data from four to six concentrations instead of only using the $LC_{50}$ value, thereby providing more statistical power. Fig 4A shows the proportion of dead individuals by the batch of mosquitoes from each collection site and time exposed to the different insecticide concentrations. Note that this figure includes data from cemeteries along the coast of Chiapas (indicated as 'panteon'); however, the generalized linear model included only collection sites within the city of Tapachula from 2016 to March 2020. We excluded coastal sites because insecticide selection pressure is unknown. In addition, we excluded bioassay data from September 2020 and 2021 because, during these years, the pyrethroid prallethrin was reintroduced by the vector control program. The analysis was run separately for permethrin and deltamethrin.

For permethrin, 625 dose-response entries were included in the model. Fig 4A shows the proportion of dead mosquitoes for each concentration, site, and time. The multivariate logistic model was significant (*p* value < 2.2e-16), with the odds of dying being explained by the three variables, insecticide concentration, the time, and the collection site (Table 1). The McFadden's Pseudo R2 for this model was 0.79 (p < 0.000). The estimates for the model (Annex 1) showed that a one-unit increase in one log of permethrin concentration (natural logarithm of concentration in μg) corresponded to an increase in the odds of dying of 814% (*p* value = 2E-16), with site and time remaining constant. One unit of time (one month) corresponded to an increase in the odds of dying by 1.5% (*p* value = 2e-16), with site and concentration remaining constant. In addition, the odds of dying were significantly explained by the collection site in 18 out of the 24 sites, holding time and concentration constant (*p* value < 2.2e-16) (Table 1). A Moran's autocorrelation analysis was conducted using the methodology described in Solis Santoyo et al (2020) [32]. For each collection period, we tested whether pyrethroid resistance ratios (RR) were explained by the geographic distance between sites. None of the analyses yielded significant correlation (p > 0.05) (S2 Table). Accordingly to a previous study conducted in Tapachula [32], lack of correlation imply that closer geographical collection sites do no share similar levels of pyrethroid resistance, and that resistance is rather explained by focal conditions at each site [32].

For deltamethrin, 656 data entries were included in the multivariate regression analysis. The three independent variables (concentration, time, and site) significantly explained the odds of dying (*p* value = 2.2e-16) (Table 1). Fig 4B shows the proportion of dead mosquitoes for each concentration, site, and time. The McFadden's Pseudo R2 for this model was 0.786 (p < 0.05). The estimates showed that a one-unit increase in one log of deltamethrin concentration corresponded to an increase in the odds of dying by 447% (*p* value = 2e-16), with time and site remaining constant. Every month after PYR removal corresponded to an increase in the odds of dying by 8.4% (*p* value = 2e-16), with site and dose remaining constant. Out of 24, 19 site estimates were significant, with six showing positive and 13 negative directions (S3 Table). Similar to permethrin bioassays, there was no spatial autocorrelation at any collection time point.

### *Kdr*-allele frequencies over time

The frequency of two knockdown resistance-associated mutations (*kdr)* in the VGSC were screened in ~50 individuals from each of the 144 collection sites from 2016 to 2020. Genotypic frequencies were calculated for 7,013 individual mosquitoes. At the V1016I locus, alleles segregated in all 144 collection sites. Meanwhile, at the F1534C locus, alleles segregated at 113 sites, mostly because of fixation of the resistant allele at 31 sites. Frequencies of the resistant alleles at loci F1534C (C1534) and V1016I (I1016) and their 95% Bayesian high-density intervals (HDI)

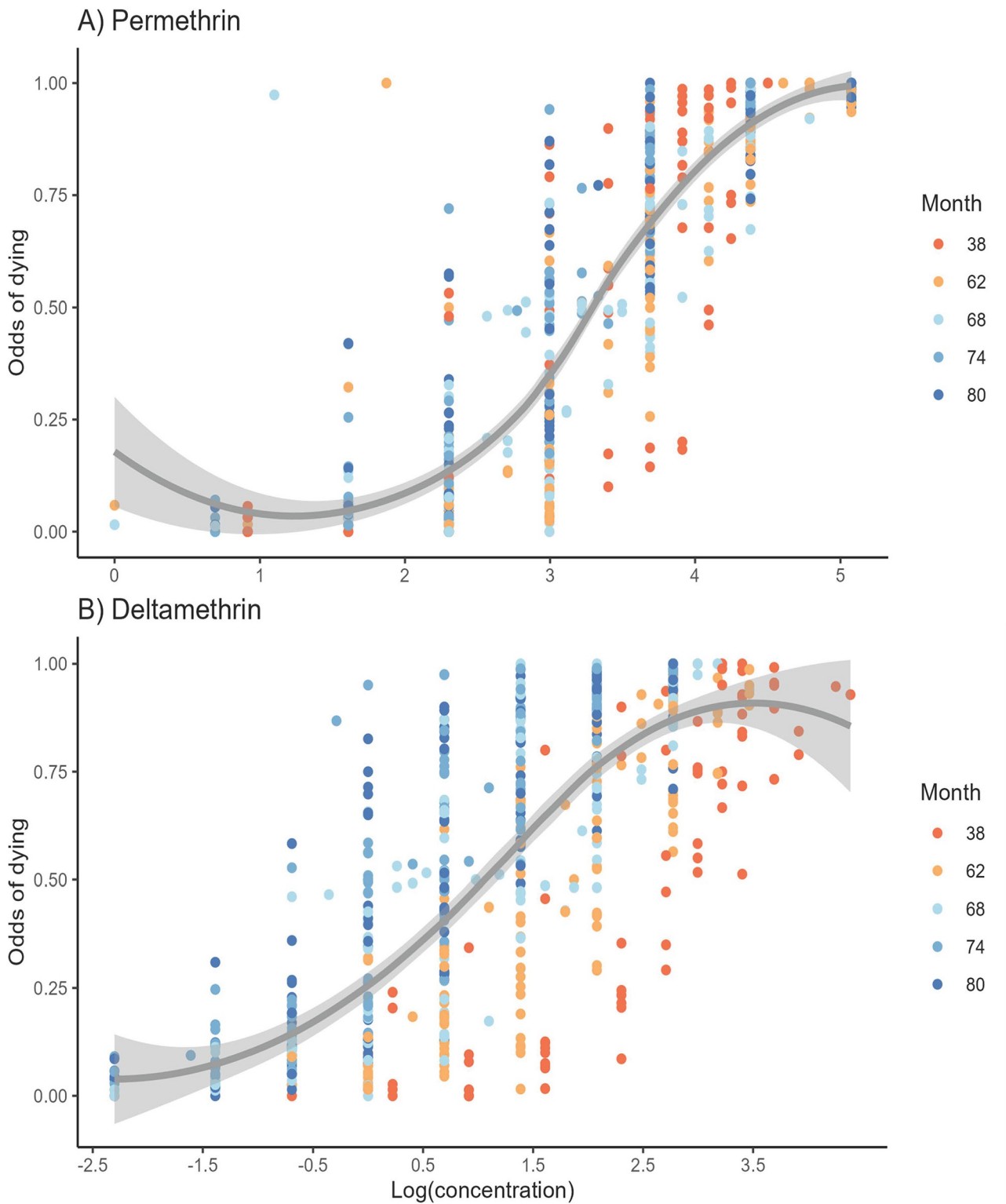

**Fig 4. Proportion of dead mosquitoes from a specific site, collection time, and concentration for:** A) Permethrin and, B) Deltamethrin. Orange-blue scale colors represent the number of months after PYRs were discontinued by vector control programs.

**Table 1. Analysis of variance for multivariate logistic regression for permethrin and deltamethrin.** The model tests the odds of dying as a response to insecticide concentration, time in months (after PYR discontinuations) and site (24 sites). The intercept, estimates and change in the odds of dying (%) for one unit of time (month) or one unit (natural log of ug/bottle) of insecticide concentration holding other factors constant is shown.

| Insecticide | Variable | ANOVA | | | GLM regression estimates | | |
|---|---|---|---|---|---|---|---|
| | | LR | Chisq | p value | Estimate | p value | Change |
| Permethrin | Intercept | | | | -7.348 | 2.00E-16 | |
| | Concentration | 26314.7 | 1 | 2.2E-16 | 2.098 | 2.00E-16 | 814% |
| | Month | 220.7 | 1 | 2.2E-16 | 0.015 | 2.00E-16 | 1.50% |
| | Site | 1336.7 | 24 | 2.2E-16 | -1.3 to 0.21 | ** | |
| Deltamethrin | Intercept | | | | -6.781 | 2.00E-16 | |
| | Concentration | 22917 | 1 | 2.2E-16 | 1.498 | 2.00E-16 | 447% |
| | Month | 5444.3 | 1 | 2.2E-16 | 0.0807 | 2.00E-16 | 8.40% |
| | Site | 1528.7 | 24 | 2.2E-16 | -0.95 to 0.6 | ‡ <0.005 | |

Df = degrees of freedom.

** 18 out of 24 site estimates were significant (<0.05), two of them were positive and 16 were negative.

‡ 10 out of 24 site estimates were significant.

are shown in Fig 5. In 2016, we screened ten sites in the city of Tapachula and seven sites along the coast of Chiapas (cemeteries). Nonoverlapping 95% HDI suggested significant differences between the city and coastal collections. Resistant I1016 frequencies were significantly higher in the city (mean = 0.48, 95% HDI = 0.45–0.507) than along the coast (mean = 0.31, HDI = 0.28–0.35). For resistant allele C1534, city frequencies were also higher (mean = 0.97, 95% HDI = 0.968–0.983) than along the coast (mean = 0.889, 95%HDI = 0.86–0.91). Because of these differences between city and coastal sites, the multivariate regression analysis included only the city collection sites from 2016 to 2020. The C1534 allele frequency was close to fixation in 2016, 2018, 2019-M, and 2019-S (mean allele frequency from 0.93 to 0.98). Then, from March to September of 2020, the mean frequency declined from 0.91 to 0.84 (95%HDI = 0.83–0.86), respectively. For I1016, no clear tendency for decline was observed from 2016 to 2019; however, in March 2020, the frequencies were significantly higher than at other collection times. These higher frequencies seemed to be a response to the reintroduction of pyrethroids in vector control programs.

**Relation between LC$_{50}$ and *kdr*-alleles.** In Mexican mosquito populations, C1534 and I1016 occurred together more often than expected [17]. In our linkage disequilibrium analysis among 144 collections, resistant alleles at C1543 and I1016 were not in disequilibrium, possibly because C1534 was close to fixation in most of the collections. To test whether a particular dilocus genotype explains the variation in LC$_{50}$s, we conducted a multivariate regression analysis using LC$_{50}$ as the response. Independent variables were site, month, and the proportion of each of the nine possible dilocus genotype combinations (Fig 6). The first two letters correspond to the 1016 locus and the second two letters to locus 1534. An individual with homozygote-susceptible genotypes at both loci (V1016I and F1534C) were designated as VVFF, whereas the double-resistant homozygote was assigned IICC. The most common dilocus genotypes among collections were IVCC, IICC, and VVCC (Fig 6). From 2016 to 2018, the frequency of the resistant IICC genotype declined, whereas VVCC increased. From March 2019 to September 2020, IICC increased again. The increase seemed to be a response to pyrethroid reintroduction in vector control programs in 2019 and 2020. Then, in September 2020, IICC decreased significantly. The model for permethrin included 74 sites with phenotypes and genotypes. The permethrin LC$_{50}$s were significantly explained by the VVCC haplotype (estimate = -1.465, T = -2.192, *p* value = 0.0337). For deltamethrin, LC$_{50}$s were only significantly explained by the month (estimate = -0.056, T = -7.42, *p* value = 2.7E-09).

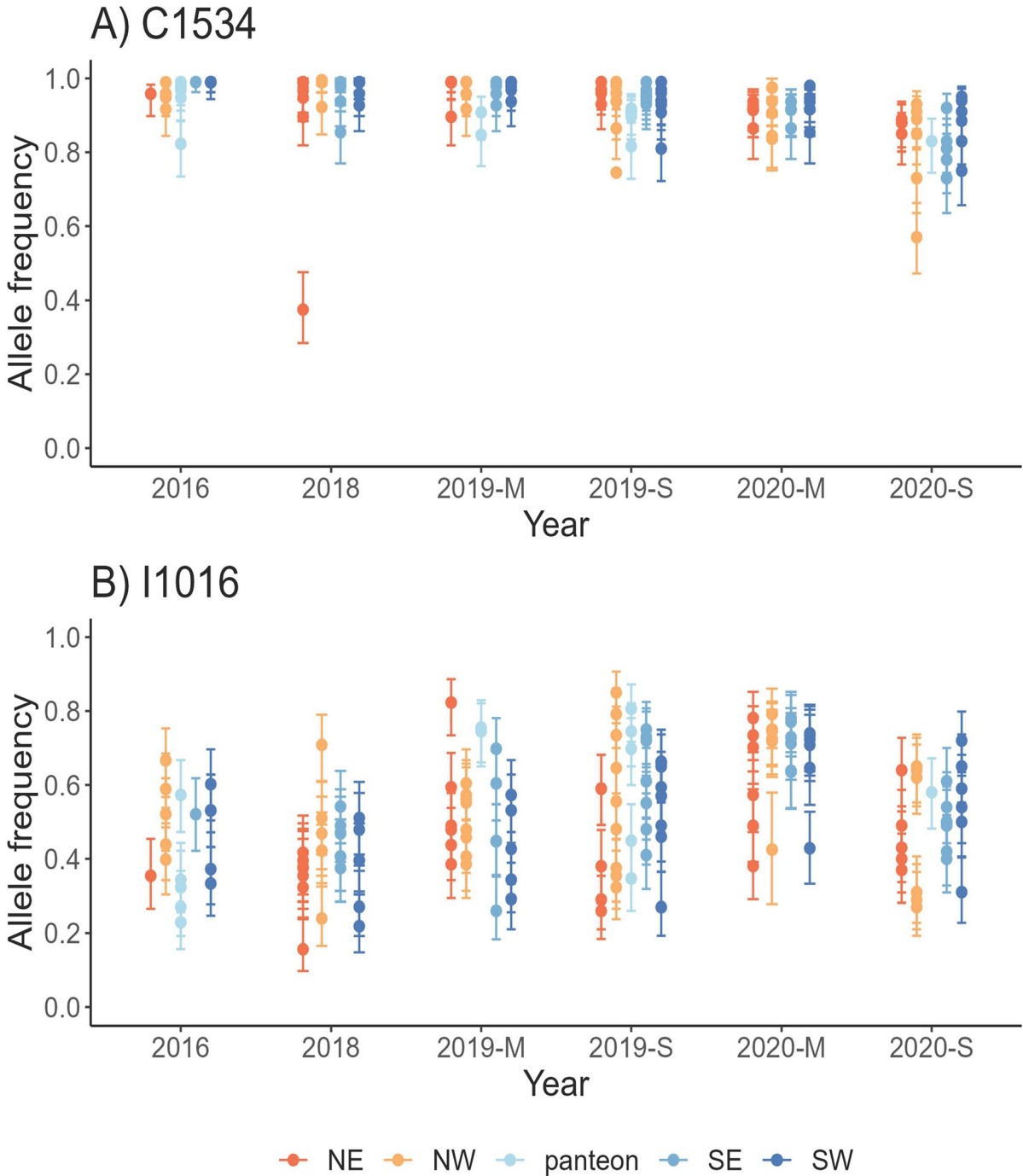

**Fig 5. Allele frequency of *kdr* mutations in *Aedes aegypti* from Tapachula and coastal Chiapas over time for: A) C1534 and B) I1016.** Approximately 50 mosquitoes were genotyped to calculate the allele frequencies at each site and time point. Sites are categorized in four geographical quadrants in Tapachula (NW = northwest, NE = northeast, SW = southwest, and SE = southeast). Collections across coastal Chiapas were obtained from cemeteries (panteon). M = March and S = September. Bars represent the 95% HDI around the frequency.

## Discussion

We tested the hypothesis that PYR resistance carries negative fitness and that susceptibility is restored in field mosquito populations when pyrethroids are not used anymore. To account

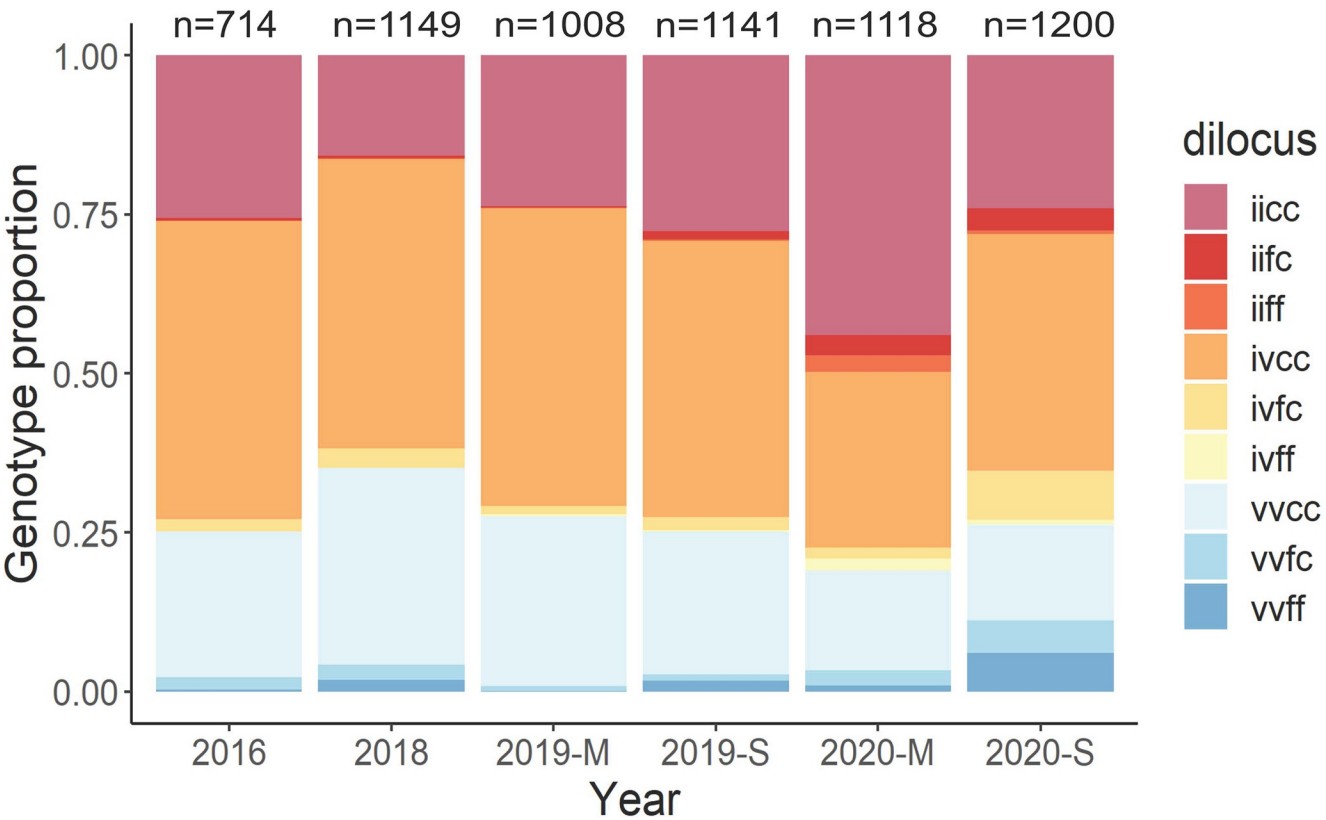

**Fig 6. Proportion of mosquitoes carrying each of the nine dilocus genotypes over 5 years of study.** The number of mosquitoes analyzed per year ranged from 714 to 1200. The first two letters correspond to the amino acid at the V1016I locus (resistant = I, susceptible = V) and the second pair corresponds to the amino acid at F1534C locus (resistant = C, susceptible = F). Dilocus genotypes go from double-resistant homozygote = IICC, to double-susceptible homozygote = VVFF.

for the large heterogeneity in field mosquito populations, we screened 24 collection sites every year in Tapachula. Our results indicate that mosquitoes from the 24 sites became more susceptible to pyrethroids over time. The model detected that with every passing month after the discontinuation of pyrethroids by vector control programs, mortality increased by 8.4% and 1.5% for deltamethrin and permethrin, respectively. This decline in resistance was also evident for one of the *kdr* mutations, for which the resistant C1534 allele declined 12% after near fixation in the initial part of our study. These results were expected, as several laboratory investigations have demonstrated that PYR resistance reversed in less than 15 generations in the absence of insecticides. However, few studies have evaluated the response of mosquito populations in the field where mosquitoes are under different insecticide selection pressures. Recently, one study in Florida measured the phenotypic and genotypic response of PYR-resistant *Ae. aegypti* populations following 2.5 years of the operational use of organophosphates [11]. The results did not detect a decline in mortality with the CDC bottle bioassay; however, the authors observed a significant decline in the frequency of double homozygous *kdr* resistant genotypes (V1016I and F1534C). In our study, however, we identified decreases in both the pyrethroid $LC_{50}$s and *kdr* genotypes. One potential explanation for these differences is that the methods used to measure resistance targeted different resistant phenotypes. The study in Florida used a time-response assay to measure knockdown in mosquitoes at 30 min of exposure. In comparison, the dose-response assay in our study measured the mortality at 24 h using different pyrethroid

concentrations. Although, the bottle bioassay has shown greater variation in the dose–response curves in comparison with the WHO tube test and the topical application bioassay [36], scoring mortality at 24 h instead of 30 min is a better predictor of resistance because it accounts for recovery due to metabolic resistance mechanisms [37]. Whether the dose-response assays are more sensitive than the time-response assays to identify changes in pyrethroid resistance is of special interest, as the interpretation of results might have important implications at the operational level.

Despite the significant decline in pyrethroid $LC_{50}$s over time, our results show that the RR levels are still higher than 10-fold almost 8 years after the discontinuation of PYRs. This finding is consistent with an observational study in Sao Paulo, Brazil, in which levels of PYR resistance remained high 10 years after the switch to OPs to control *Ae. aegypti* [38]. The authors suggested that pyrethroid resistance remained high in mosquito populations because of the widespread use of pyrethroids in consumer-based products for domestic and residential application. For instance, a survey in the city of Merida, Mexico, showed that 87% of households regularly use pyrethroid-based commercial products to reduce mosquito nuisance [39]. Moreover, the use of PYR surface sprays led to a significant increase in the frequency of V1016I *kdr* homozygotes in surviving *Ae. aegypti*, suggesting strong selection pressure for this allele [40]. Strikingly, even when our study did not account for the use of pyrethroid-based commercial products and despite the large heterogeneity in PYR resistance across the 24 sites, our model was able to identify a decline in PYR resistance over time. A limitation in our model is that we cannot predict the time required to reach lower levels of resistance ($< 5$-fold), which would require continuing to monitor for resistance for several years. Another limitation is that our surveillance began 3 years after the discontinuation of the PYRs, and we lack information about the PYR resistance levels during the peak of PYR use in Tapachula. Our closest references are three studies that evaluated *Ae. aegypti* PYR resistance in Mexico using the same methods. In 2007, five collection sites in the states of Quintana Roo and Yucatan showed permethrin RRs ranging between 2.6- and 10.2-fold [24]. In 2009, seven sites in the state of Veracruz showed deltamethrin RRs were lower than 3-fold at four sites and between 16- and 20-fold at three sites. In the same study, permethrin RRs were lower than 10-fold at four sites and between 11- and 33-fold at three sites [13]. In 2014, eight sites from the states of Yucatan, Guerrero, and Chiapas showed deltamethrin RRs ranging between 18- and 108-fold whereas permethrin RRs ranged between 15- and 60-fold. In this last study, Tapachula showed RRs of 25- and 95-fold for permethrin and deltamethrin, respectively [29]. These findings suggest high levels of pyrethroid RRs were widespread just before our study. However, because large heterogeneity in pyrethroid RRs can be found at the city scale [32], making assumptions about resistance levels across states and different studies should be done with caution.

Interestingly, our RRs were higher for deltamethrin (PYR type-2) than for permethrin (PYR type-1). This finding was unexpected, as mosquito populations were subject to PYR type-1 rather than PYR type-2 from 1999 to 2013. One possible explanation could be that mosquito populations were exposed to PYR type-2 selection from the use of consumer-based pyrethroid products. Future studies should evaluate the use of these products at the study sites. A second observation was that PYR discontinuation by vector control programs resulted in higher deltamethrin decline rates than those rates observed for permethrin; however, *kdr* dilocus genotypes did not explain the variation. These results suggest a possible higher fitness cost for deltamethrin resistance in the absence of insecticides. Specific mechanisms of resistance for pyrethroids type-1 and type-2 might include a combination of several mechanisms such as specific *kdr* mutations, compensatory ion channel mutations, cuticle modification, and specific enzyme detoxifying variants with different fitness costs in mosquitoes. However, a recent study showed that metabolic resistance was associated with higher fitness costs than target-site

resistance in the absence of insecticides [41]. From the three congenic *Ae. aegypti* strains carrying either CYP-mediated, *kdr*-mediated, or CYP + *kdr* mechanisms, strains with CYP-mediated resistance showed a significantly reduced net reproductive rate relative to the susceptible and the *kdr* strains [41]. In our study, we observed a major decline in PYR RRs from 2016 to 2019, which did not match the decline in *kdr* alleles that occurred mostly from 2019 to 2020. We hypothesize that metabolic resistance mechanisms or epistatic factors might have had a greater impact on the resistance levels from 2016 to 2019. A limitation of our study is that we did not test the change of detoxification mechanisms. However, a parallel study by our team measured the enzymatic activity of CCE, CYP, and GST in collections taken in 2018 and 2020 from 22 sites in Tapachula [42]. The number of sites with mean enzymatic activity higher than the susceptible strain (NO) declined for alpha-, beta-, and PNPA- esterases at three, one, and five collection sites, respectively. In addition, an increase was observed in the number of sites with higher GST and CYP activity than the susceptible strain. The enzyme activity levels were not significantly associated with PYR RRs, but bioassays using PYRs in combination with enzyme inhibitors significantly increased the mortality, suggesting biochemical assays might not be as sensitive as synergist bioassays to explain the role of metabolic resistance in mosquito populations. Including DNA or RNA markers of insecticide detoxification genes might be a good option to explore the role of metabolic resistance in mosquito populations. However, this requires a fine understanding of the genes conferring resistance to a particular insecticide in a geographical region and their variation within and between populations [24]. For instance, single nucleotide polymorphisms (SNPs) in detoxification genes have been associated with PYR resistance in our study sites, and we are in the process of testing their segregation and association with different levels of resistance in field mosquitoes [43].

Many studies have intended to use *kdr* frequencies as a proxy for pyrethroid resistance. Particularly in this study, *kdr* mutations explained somewhat the large variation among pyrethroid RRs. Only one dilocus genotype (VVCC) significantly explained the large variation in permethrin $LC_{50}$s; however, no *kdr* dilocus genotype explained deltamethrin $LC_{50}$s. From 2016 to 2019, C1534 allele frequencies had low spatial and temporal heterogeneity because of proximity to fixation, whereas I1016 frequencies had large spatial heterogeneity within a 0.35 to 0.68 range. The lack of association between the PYR $LC_{50}$s and *kdr*-dilocus genotypes could be because the method to measure resistance in our study (24-h mortality) was not only measuring *kdr* but also was measuring recovery mechanisms due to metabolic resistance. Recovery rates among knocked-down mosquitoes ranged between 30 and 45% at our study sites. A previous genome-wide association mapping from pools of mosquitoes exposed to a permethrin $LC_{50}$ showed that mosquitoes exhibiting knockdown resistance at 1 h of exposure were strongly associated with *kdr* mutations (V410L and V1016I), whereas mosquitos recovering at 4 h were associated with SNPs at a different group of ion channels and detoxifying enzyme-coding genes [44]. The operational implication of mosquito recovery has not been evaluated in the field, and its role in selection of pyrethroid resistance mechanisms might be underestimated.

One important observation in our study is that significant changes in *kdr* frequencies occurred between 2020 and 2021, with C1534 decreasing by 6%, whereas I1016 increased by 15%. The increase in I1016 frequencies occurred simultaneously with an increase in pyrethroid $LC_{50}$s, following the use of imidacloprid and prallethrin (PYR type-1) during whole-city spraying and focal indoor spatial spraying from 2019 to 2020. Such a rapid increase was also documented in Iquitos, Peru, where *kdr* frequencies and heterogeneity increased following a 6-week cypermethrin spray application in 2013 and 2014 [45]. The rapid response in resistance indicated a strong impact of vector control activities on the susceptibility of mosquito populations. Similarly, our study illustrates how short periods of pyrethroid use offset the trends in PYR resistance reversal. Inevitably, IRM strategies will require long-term commitments

between vector control programs and community participation to manage the use of pyrethroids and allow the natural course of pyrethroid reversal in resistant populations over time.

## Conclusion

The trends in insecticide resistance in our study followed the application of insecticides by vector control programs, suggesting that operational control seems to have a significant impact in resistance levels across the mosquito collections. Our results in the field support the hypothesis that PYR resistance has negative fitness in the absence of pyrethroids. We observed a decline in PYR resistance over time despite high heterogeneity, heavy application of OPs, and possibly residential-level use of consumer-based PYR products. Despite the significant decline in the range of PYR RRs, resistance is still considered high. Whether the PYR RRs calculated in the laboratory are associated with differences in operational effectiveness requires further evaluation. Our results suggest that pyrethroid reversal is possible; however, long-term IRM by vector control programs and community participation are needed to speed the process.

## Disclaimer

The findings and conclusions in this report are those of the authors and do not necessarily represent the official position of CDC.

## Supporting information

**S1 Table. Location of *Aedes aegypti* cemetery collections in the coast of Chiapas.** Larvae were collected from each cemetery on 2016's wet season. The municipality name and geographical coordinates are provided.
(DOCX)

**S2 Table. Moran's I statistic showing autocorrelation coefficients between the pyrethroid RR's and the geographic distance between 24 *Aedes aegypti* collection sites in Tapachula, Mexico.** RR's were obtained by bottle bioassays. Analyses were ran separately for each pyrethroid (permethrin and deltamethrin) and each collection time point, including 2018, 2019, and 2020. S = September and M = March. *p* values above 0.05 are considered non-significant. N = the number of pairwise comparisons between sites in the specific distance class.
(DOCX)

**S3 Table. Summary of generalized linear model estimates for A) Permethrin and, B) Deltamethrin.** Independent variables include the insecticide concentration (log(concentration)), site (24 sites), time in months (after PYR discontinuation). Estimates, standard error, z-value, probability and change are shown for each variable.
(DOCX)

## Acknowledgments

We thank Francisco Pozos and Eduardo Vazquez Lopez from the Vector Control Program in Tapachula District VII for sharing information about insecticide use and coordinating larval collections during this study. We acknowledge the technical support received in the field and the laboratory of the Insecticide Resistance Group at CRISP.

## Author Contributions

**Conceptualization:** Patricia Penilla-Navarro, Alma Lopez-Solis, Americo D. Rodriguez, William C. Black IV, Karla Saavedra-Rodriguez.

**Data curation:** Alma Lopez-Solis, Farah Vera-Maloof, Saul Lozano, Karla Saavedra-Rodriguez.

**Formal analysis:** Farah Vera-Maloof, Saul Lozano, Karla Saavedra-Rodriguez.

**Funding acquisition:** Americo D. Rodriguez, William C. Black IV.

**Investigation:** Francisco Solis-Santoyo, Alma Lopez-Solis, Americo D. Rodriguez, Elsa Contreras-Mejía, Geovanni Vázquez-Samayoa, William C. Black IV.

**Methodology:** Francisco Solis-Santoyo, Alma Lopez-Solis, Americo D. Rodriguez, Rushika Perera.

**Project administration:** Patricia Penilla-Navarro, Francisco Solis-Santoyo, Alma Lopez-Solis, Americo D. Rodriguez, William C. Black IV, Karla Saavedra-Rodriguez.

**Resources:** Americo D. Rodriguez, Elsa Contreras-Mejía, Geovanni Vázquez-Samayoa, Rene Torreblanca-Lopez, Rushika Perera.

**Software:** Saul Lozano.

**Supervision:** Patricia Penilla-Navarro, Francisco Solis-Santoyo, Alma Lopez-Solis, Americo D. Rodriguez, William C. Black IV, Karla Saavedra-Rodriguez.

**Validation:** Patricia Penilla-Navarro, Francisco Solis-Santoyo, Alma Lopez-Solis, Farah Vera-Maloof, Saul Lozano.

**Visualization:** Karla Saavedra-Rodriguez.

**Writing – original draft:** Karla Saavedra-Rodriguez.

**Writing – review & editing:** Patricia Penilla-Navarro, Farah Vera-Maloof, Saul Lozano, Karla Saavedra-Rodriguez.

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
