## [Decision Letter · Decision Letter 0]

25 Jul 2023

Dear Dr. Saavedra-Rodriguez,

Thank you very much for submitting your manuscript "Pyrethroid susceptibility reversal in Aedes aegypti: a longitudinal study in Tapachula, Mexico" for consideration at PLOS Neglected Tropical Diseases. As with all papers reviewed by the journal, your manuscript was reviewed by members of the editorial board and by several independent reviewers. In light of the reviews (below this email), we would like to invite the resubmission of a significantly-revised version that takes into account the reviewers' comments. 

The reviewers acknowledged the work's relevance and worthness of publication. However, important issues were raised, in particular by Reviewer #3, whom questioned why the reference strain was not used in all experiments and requested additional analysis to demonstrate the doses tested were appropriate.

We cannot make any decision about publication until we have seen the revised manuscript and your response to the reviewers' comments. Your revised manuscript is also likely to be sent to reviewers for further evaluation.

Sincerely,

Tereza Magalhaes, Ph.D.

Academic Editor

Álvaro Acosta-Serrano

Section Editor

The reviewers acknowledged the work's relevance and worthness of publication. However, important issues were raised, in particular by Reviewer #3, whom questioned why the reference strain was not used in all experiments and requested additional analysis to demonstrate the doses tested were appropriate.

Reviewer's Responses to Questions

**Key Review Criteria Required for Acceptance?**

**Methods**

-Are the objectives of the study clearly articulated with a clear testable hypothesis stated?

-Is the study design appropriate to address the stated objectives?

-Is the population clearly described and appropriate for the hypothesis being tested?

-Is the sample size sufficient to ensure adequate power to address the hypothesis being tested?

-Were correct statistical analysis used to support conclusions?

-Are there concerns about ethical or regulatory requirements being met?

Reviewer #1: -Number of replicates for each assay was not stated. 

-State the total number of mosquitoes used in bioassay and kdr genotyping

-F1/F2 mosquitoes were used. How did you give a blood meal to the mosquitoes?

Reviewer #2: This is in-depth longitudinal study on the Pyrethroid resistant level in Aedes aegypti using an IRM Strategy. It's hypotheses are clearly stated, the sample size adequate and methods are appropriate for the study.

Reviewer #3: There are items that are used in analysis that are not described in the methods, particularly collection sites. There is also confusion on what some terms are referring to. In particular concentration of insecticide. Does it refer to assay or field application on page 9. 

When running assays that calculate RR the comparator (Rock strain in this study) should ALWAYS be run in real time with test populations. Not periodically.

**Results**

-Does the analysis presented match the analysis plan?

-Are the results clearly and completely presented?

-Are the figures (Tables, Images) of sufficient quality for clarity?

Reviewer #1: Appropriate

Reviewer #2: Graphs could benefit with some indication of significant results but otherwise the results are clearly presented and sufficient.

Reviewer #3: There is a duplicate figure. Figures 5 and Figure S1 are the same. The only difference is 5 is separated into 2 panels and S1 combines them in 1 graph. Other comments are in uploaded document. 

Looking for geographic trends was hinted at but not clearly described.

Chi squares for LC50 calculations need to be added.

**Conclusions**

-Are the conclusions supported by the data presented?

-Are the limitations of analysis clearly described?

-Do the authors discuss how these data can be helpful to advance our understanding of the topic under study?

-Is public health relevance addressed?

Reviewer #1: Detoxification mechanism was not studied and this limitation must be acknowledged.

Reviewer #2: The discussion adequately addresses the limitation and relevance of the study. It is a well written and logical discussion and conclusion.

Reviewer #3: Conclusions are well written.

**Editorial and Data Presentation Modifications?**

Reviewer #1: n/a

Reviewer #2: (No Response)

Reviewer #3: see attachment

**Summary and General Comments**

Reviewer #1: This is an interesting study and the findings could be useful in developing effective control measures against dengue. Prior to acceptance, please illustrate the method in details so that it can be replicated by the other researchers. Detoxification is the main resistance mechanism in Aedes but this was not investigated. Please acknowledge this limitation.

Reviewer #2: Overall, A well written and logical paper, highly valuable for the assessment of applied IRM strategies.

Reviewer #3: see attachment

PLOS authors have the option to publish the peer review history of their article (what does this mean?). If published, this will include your full peer review and any attached files.

Reviewer #1: No

Reviewer #2: No

Reviewer #3: No
---

## [Editor Report · Decision Letter 1]

27 Nov 2023

Dear PhD Saavedra-Rodriguez,

We are pleased to inform you that your manuscript 'Pyrethroid susceptibility reversal in Aedes aegypti: a longitudinal study in Tapachula, Mexico' has been provisionally accepted for publication in PLOS Neglected Tropical Diseases.

Best regards,

Tereza Magalhaes, Ph.D.

Academic Editor

Álvaro Acosta-Serrano

Section Editor

---

## [Editor Report · Acceptance letter]

21 Dec 2023

Dear PhD Saavedra-Rodriguez,

We are delighted to inform you that your manuscript, "Pyrethroid susceptibility reversal in *Aedes aegypti*: a longitudinal study in Tapachula, Mexico," has been formally accepted for publication in PLOS Neglected Tropical Diseases.

Best regards,

Shaden Kamhawi

co-Editor-in-Chief

Paul Brindley

co-Editor-in-Chief
